# Ruthenium anchored on carbon nanotube electrocatalyst for hydrogen production with enhanced Faradaic efficiency

Do Hyung Kweon [1], Mahmut Sait Okyay [2], Seok-Jin Kim [1], Jong-Pil Jeon[1], Hyuk-Jun Noh[1], Noejung Park [2], Javeed Mahmood [1✉] & Jong-Beom Baek [1✉]

Developing efficient and stable electrocatalysts is crucial for the electrochemical production of pure and clean hydrogen. For practical applications, an economical and facile method of producing catalysts for the hydrogen evolution reaction (HER) is essential. Here, we report ruthenium (Ru) nanoparticles uniformly deposited on multi-walled carbon nanotubes (MWCNTs) as an efficient HER catalyst. The catalyst exhibits the small overpotentials of 13 and 17 mV at a current density of 10 mA cm$^{-2}$ in 0.5 M aq. H$_2$SO$_4$ and 1.0 M aq. KOH, respectively, surpassing the commercial Pt/C (16 mV and 33 mV). Moreover, the catalyst has excellent stability in both media, showing almost "zeroloss" during cycling. In a real device, the catalyst produces 15.4% more hydrogen per power consumed, and shows a higher Faradaic efficiency (92.28%) than the benchmark Pt/C (85.97%). Density functional theory calculations suggest that Ru–C bonding is the most plausible active site for the HER.

[1] School of Energy and Chemical Engineering / Center for Dimension-Controllable Organic Frameworks Ulsan National Institute of Science and Technology (UNIST), 50 UNIST, Ulsan 44919, South Korea. [2] School of Natural Science Ulsan National Institute of Science and Technology (UNIST), 50 UNIST, Ulsan 44919, South Korea. ✉email: javeed@unist.ac.kr; jbbaek@unist.ac.kr

Given the ongoing depletion of fossil fuels and growing global environmental challenges, the search for carbon less (or free) energy is taking on increasing importance in energy engineering. Among carbon-free energy sources, hydrogen ($H_2$), is particularly popular because it contributes no environmental pollutants[1]. The most promising eco-friendly and economical way to produce pure hydrogen is by electrochemical water splitting[2–5]. To ensure the hydrogen evolution reaction (HER) is efficient and continuous, the catalyst must promote proton reduction with minimal overpotential, to minimize additional energy consumption[6,7]. This requirement has made the efficient production of hydrogen using electrochemical catalysts a challenge for scientists over the last several decades[8–14].

Platinum (Pt) is still considered the benchmark catalyst for the HER, with low overpotentials, small Tafel slopes and high exchange current densities due to its optimum binding force with hydrogen[15]. However, in addition to soaring cost and scarcity, Pt has poor electrochemical stability, which is associated with leaching in corrosive electrolytes and irreversible aggregation of Pt nanoparticles by Ostwald ripening[16,17], limiting its practical applications. In order to replace Pt, efforts have been devoted to developing earth abundant element-based catalysts for HER, e.g., phosphates[18], carbides[19,20], oxides[21], and transition metal sulfides[15,22]. However, they typically suffer from both limited electrochemical activity and durability.

Recent efforts have focused on designing new catalysts with superior activity and durability compared to commercial Pt[23–25]. Among the many metal-based catalysts evaluated for HER catalysis, ruthenium (Ru), one of the platinum group metals, has been widely tested, because of its low-cost (1/3 the price of Pt)[26], high HER efficiency, and stability[25,27,28]. In principle, HER efficiency is closely related to the strength of the metal-hydrogen (M–H) bonds on the surface of the catalysts[29–37] and the overpotential required for hydrogen reduction. The Gibbs free energy ($\Delta G_H$) of the Ru–H bond is very close to that of the optimum Pt–H bond at the center of the volcanic plot for HER[25,38,39]. But even though Ru has potentially high electrochemical HER activity, it is prone to agglomerate, because it has a much larger cohesive energy than Pt[40]. To resolve this issue, a strategy of uniformly dispersing and sequestering Ru nanoparticles in a two-dimensional (2D) carbon structure was developed, and it demonstrated excellent HER performance with low overpotentials, outstanding durability and high turnover frequencies in both acidic and alkaline conditions[25].

Developing methods to produce active but low-cost catalysts remains one of the most crucial obstacles to the realization of a hydrogen economy. Among various approaches, carbon-based materials have attracted interest as low-cost supports for active HER catalysts. Various advanced electrocatalysts have been fabricated by incorporating electrochemically active transition metals into one- or two-dimensional carbon nanostructures, including carbon nanotubes[41] and graphene nanosheets[42]. These conductive supports are important because they enable the mass production of highly efficient and stable catalysts at low-cost. And in addition to the activity of the catalytic metal nanoparticles, the conductive supports can also make a significant contribution to the overall catalytic performance. For efficient catalysis, the catalytic nanoparticles need to be dispersed and stabilized on an appropriate substrate.

Here, we demonstrate that an electrocatalyst of Ru nanoparticles anchored on multiwalled carbon nanotube (Ru@MWCNT) is capable of catalysing HER with excellent activity and stability. The Ru@MWCNT catalyst exhibits superior HER activity to Ru@MWCNT and commercial Pt/C catalysts in both acidic and alkaline media. Notably, Ru carboxylate complex is formed through the introduction of carboxylic acid groups (–COOH) on MWCNT to form uniform and small Ru nanoparticles. This suggests the formation of Ru nanoparticles, Ru–C and Ru–O bonds through Extended X-ray absorption fine structure (EXAFS). In the actual water-splitting system construction and analysis, Ru@MWCNT produces 15.4% more hydrogen per power consumption than commercial Pt/C and Faradaic efficiency (92.28%) is higher than Pt/C (85.97%). Density functional theory (DFT) calculations identify the Ru–C structure as the most plausible active site structure with most stable energies for hydrogen binding energies of possible H binding sites. The Ru@MWCNT catalysts comprising Ru–C sites as reported herein have appropriate hydrogen binding energies for HER, and strong Ru–C bonding energies reflects the excellent stability.

## Results

**Preparation and characterization of catalyst.** A simple schematic diagram of the Ru nanoparticle-impregnated MWCNT (Ru@MWCNT) catalyst is shown in Fig. 1. Commercial MWCNTs were mildly oxidized with nitric acid to introduce oxygenated functional groups (specifically, carboxylic acids, –COOH) on the surface of MWCNT. With abundant carboxylic acids on the surface of the MWCNT, the Ru ions ($Ru^{3+}$) can be easily adsorbed on the surface of the MWCNT, by forming Ru carboxylate complexes[43]. Individual $Ru^{3+}$ ions were then directly reduced to $Ru^0$ nanoparticles in the presence of sodium borohydride ($NaBH_4$) to form Ru@MWCNT. Subsequent heat-treatment (thermal reduction) under inert conditions further reduced the Ru nanoparticles and oxygenated groups for improved HER performance. Extended X-ray absorption fine structure (EXAFS) spectroscopy was used to analyze the formation of Ru carboxylate complex and local structural environment of Ru@MWCNT catalyst before and after heat-treatment (Supplementary Fig. 1). As a reference, Ru acetylacetonate, containing pristine Ru–O bonds was used to confirm Ru–O bonding. The Fourier-transformed (FT) $k^2$-weighted EXAFS spectrum of the reference Ru acetylacetonate exhibits the major peak at around 1.5 Å, corresponding to Ru–O coordination. Ru@MWCNT before heat-treatment also has Ru–O coordination, which con-

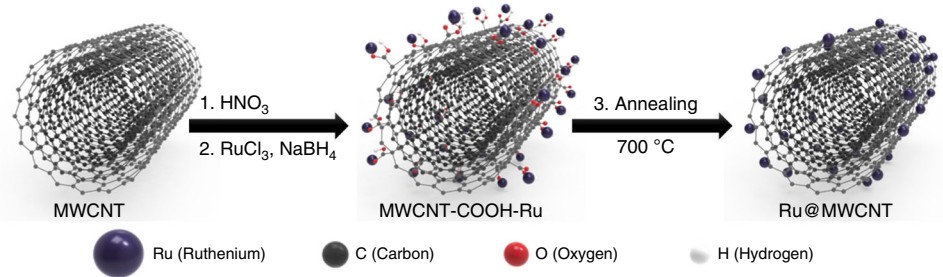

**Fig. 1** Schematic illustration of the process steps for forming Ru@MWCNT catalyst.

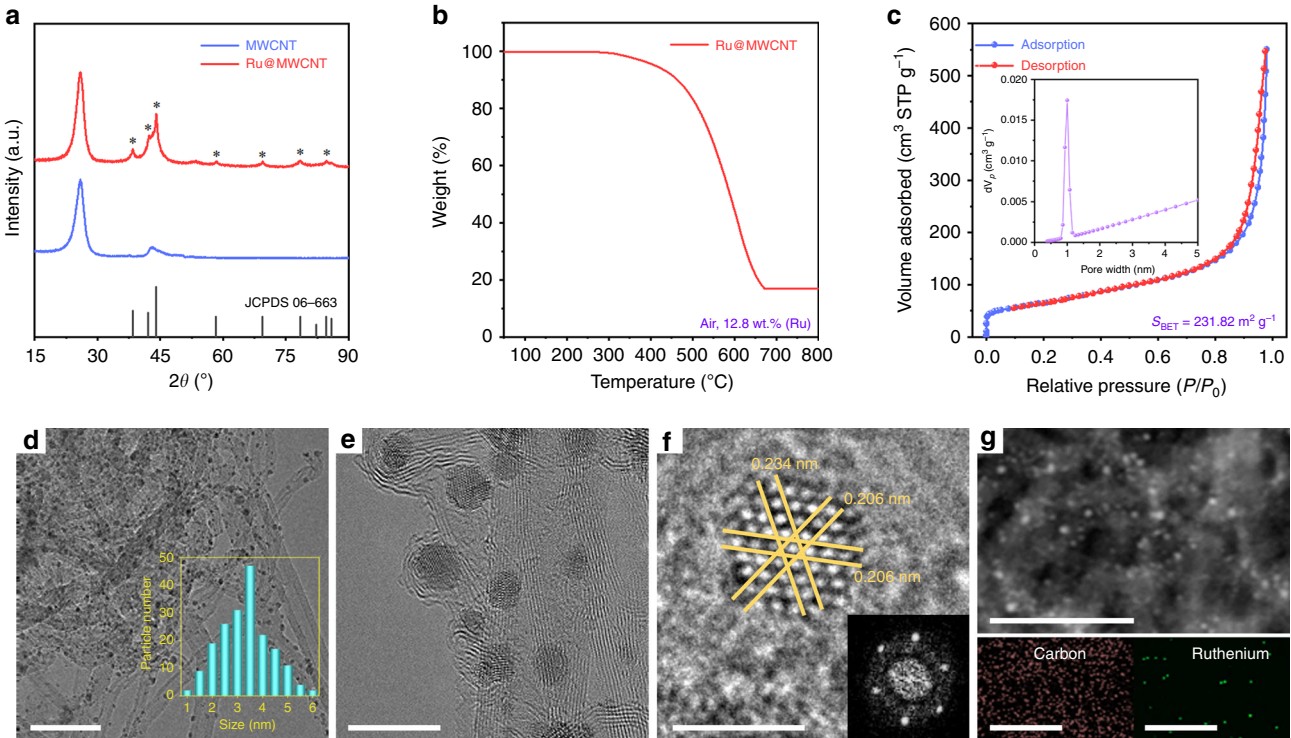

**Fig. 2 Structural and morphological characterization of the Ru@MWCNT. a** Powder XRD pattern. Asterisks in Ru@MWCNT denote Ru crystal (JCPDS 06-663). **b** TGA curve under air atmosphere at a ramping rate of 10 °C min⁻¹. **c** Nitrogen ($N_2$) adsorption–desorption isotherms at 77 K. The inset in **c** shows the pore size distribution. The specific surface area was calculated using the BET method. **d**, **e** TEM images. The inset in (**d**) shows the size distribution of Ru nanoparticles. **f** The HR-TEM image is focused on an Ru nanoparticle, showing the high crystallinity of the Ru elements, and their compact packing in the lattice. The inset in **f** is the corresponding FFT pattern. **g** High-angle annular dark-field scanning transmission electron microscopy (HAADF-STEM) image and scanning transmission electron microscopy coupled energy-dispersive X-ray spectroscopy (STEM-EDS) element mapping of Ru@MWCNT. Scale bar: **d** 50 nm; **e** 10 nm; **f** 2 nm; **g** 100 nm.

firms the Ru carboxylate coordination. However, after heat-treatment, Ru@MWCNT shows that the peak at 1.5 Å was slightly shifted to 1.6 Å, indicating the formation of Ru–C coordination[44]. The main peak at 2.4 Å is associated with Ru–Ru coordination in Ru nanoparticles[44]. These results indicate the formation of Ru carboxylate complexes, which help to form the smaller and more uniform Ru nanoparticles during the heat-treatment. To determine the optimum conditions, the Ru@MWCNT samples were heat-treated at different temperatures. The sample annealed at 700 °C showed the best HER catalytic performance in both acid and alkaline electrolytes (Supplementary Fig. 2).

The crystal structure of the Ru@MWCNT was analyzed using a high-power X-ray diffraction (HP-XRD) pattern (Fig. 2a). The peak observed at 25.6° belongs to the (002) plane of the MWCNT. The other peaks at 38.5, 42.2, 44.1, 58.4 and 69.6° can be assigned to the (100), (002), (101), (102), and (110) planes of the hexagonal Ru crystals. The average size of the Ru nanoparticles on the Ru@MWCNT was calculated to be 3.4 nm using the Scherrer equation. X-ray photoelectron spectroscopy (XPS) was used to analyze the chemical composition of the Ru@MWCNT (Supplementary Fig. 3). In the high-resolution C 1 s spectrum, the peak at 284.6 eV is associated with the graphitic C–C bonds of the MWCNT. The peak at 280.4 eV is related to the atomic state of the Ru⁰ species in the Ru@MWCNT. The bulk Ru content of Ru@MWCNT was determined by thermogravimetric analysis (TGA) in air, and was ~12.8 wt% (Fig. 2b). The value is in good accordance with the elemental analysis (Supplementary Table 1). The nitrogen ($N_2$) adsorption–desorption isotherm was obtained to calculate the specific surface area ($S_{BET}$) using the Brunauer-

Emmett-Teller (BET) method. The $S_{BET}$ of the Ru@MWCNT was found to be 231.82 m² g⁻¹ (Fig. 2c). Considering the high specific surface area and small Ru nanoparticles, the Ru@MWCNT catalyst was expected to display good HER performance.

The morphology of the Ru@MWCNT was explored by field emission scanning electron microscope (FE-SEM) and transmission electron microscopy and (TEM). The SEM images of the Ru@MWCNT revealed a clean and smooth surface morphology (Supplementary Fig. 4). The TEM images of the Ru@MWCNT clearly confirmed that the Ru nanoparticles were uniformly anchored to the surface of the MWCNT. The particle size distribution was in the range of 2–5 nm and the average size was 3.4 nm (Fig. 2d, e and Supplementary Fig. 5). Due to small particle size, and the uniform and narrow particle size distribution, a large number of Ru active sites are likely to be exposed, while the MWCNT provides an efficient electron pathway. High-resolution TEM images of the single Ru nanoparticle and the corresponding fast Fourier transform (FFT) pattern showed that the Ru elements were compactly packed into the hexagonal lattice (Fig. 2f)[25], which precisely agreed with the XRD pattern (Fig. 2a). The uniform distribution of Ru nanoparticles on the surface of the MWCNT was further confirmed by scanning transmission electron microscopy (STEM) image and corresponding energy-dispersive X-ray spectroscopy (EDS) elemental mapping images (Fig. 2g).

**Electrochemical HER activity and stability of Ru@MWCNT catalyst.** The Ru@MWCNT catalyst was evaluated for electrochemical HER performance in a $N_2$-saturated 0.5 M aq. $H_2SO_4$

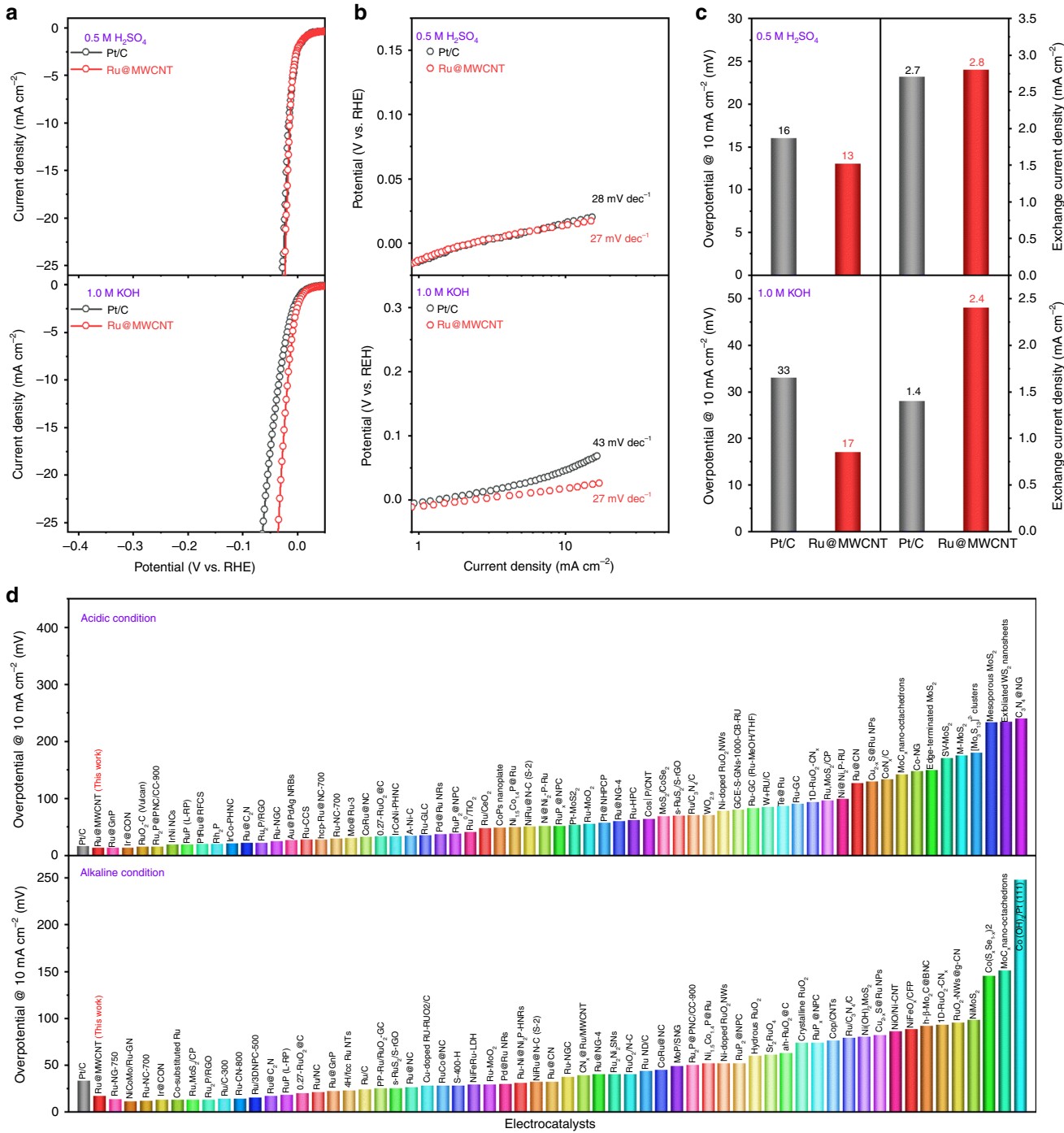

**Fig. 3 Electrochemical HER performance of the Ru@MWCNT and Pt/C catalysts. a, b** Polarization curves and corresponding Tafel plots in $N_2$-saturated 0.5 M aq. $H_2SO_4$ solution and 1.0 M aq. KOH solution. Scan rate: 5 mV s$^{-1}$. **c** Overpotentials at 10 mA cm$^{-2}$ and exchange current density in $N_2$-saturated 0.5 M aq. $H_2SO_4$ solution and 1.0 M aq. KOH solution. **d** Comparison of the overpotentials at 10 mA cm$^{-2}$ with recently reported HER catalysts in both acidic and alkaline conditions.

solution. As references, commercial Pt/C and bare MWCNT were also tested under the same conditions and compared. The MWCNT did not show catalytic activity toward HER in the range of applied potential. On the other hand, both the Pt/C and Ru@MWCNT required an overpotential of ~0 mV to induce hydrogen evolution (Fig. 3a).

Notably, the HER current density of Ru@MWCNT sharply increased as the overportential increased, with a Tafel slope of 27 mV dec$^{-1}$ similar to Pt/C (Fig. 3b). The small Tafel slope

indicates that the rate determining step is the recombination of chemisorbed hydrogen, following the Volmer-Tafel mechanism[45–47].

As a critical parameter for practical evaluation, the overpotential at a current density of 10 mA cm$^{-2}$ was evaluated for each catalyst. Ru@MWCNT displayed an overpotential of 13 mV and Pt/C required 16 mV to deliver a current density of 10 mA cm$^{-2}$. From the Tafel slope, the exchange current density of Ru@MWCNT was 2.8 mA cm$^{-2}$, which was similar to the Pt/C

(2.7 mA cm$^{-2}$), indicating Ru@MWCNT electrode's rapid HER kinetics (Fig. 3c). In acidic conditions, electrochemical impedance spectroscopy (EIS) analysis of the Ru@MWCNT catalyst exhibited a charge transfer resistance of 1.81 $\Omega$ cm$^2$ at an overpotential of 35 mV, which was lower than the Pt/C (2.23 $\Omega$ cm$^2$ at 35 mV). This implies fast electron/proton transfer at the interface of the Ru@MWCNT and the electrolyte (Supplementary Fig. 6). This remarkably improved HER performance is believed to be due to favorable charge transfer between the active sites and the working electrode, which is attributed to the highly conductive MWCNT substrate.

The HER efficiencies of the Ru@MWCNT and commercial Pt/C catalysts were evaluated in N$_2$-saturated 1.0 M aq. KOH solution (Fig. 3a). Interestingly, the Ru@MWCNT catalyst exhibited a smaller Tafel slope of 27 mV dec$^{-1}$ than the Pt/C (43 mV dec$^{-1}$). The smaller Tafel slope indicates that Ru@MWCNT catalyzed the reaction faster than Pt/C (Fig. 3b). The exchange current density (2.4 mA cm$^{-2}$) of the Ru@MWCNT was also higher than the Pt/C (1.4 mA cm$^{-2}$), indicating it had higher electrocatalytic HER activity in alkaline medium (Fig. 3c). As a result, the overpotential required to generate a current density of 10 mA cm$^{-2}$ was only 17 mV, smaller than the benchmark Pt/C (33 mV). The charge transfer resistance of the Ru@MWCNT calculated from EIS was 2.38 $\Omega$ cm$^2$ at an overpotential of 45 mV, while that of Pt/C was 4.22 $\Omega$ cm$^2$ (Supplementary Fig. 6). The lower charge transfer resistance of Ru@MWCNT also indicates efficient HER charge transfer kinetics compared to Pt/C in alkaline conditions.

The overpotentials of Ru@MWCNT at 10 mA cm$^{-2}$ in acidic (Fig. 3d and Supplementary Table 2) and alkaline media (Fig. 3d and Supplementary Table 3) were compared with other HER catalysts reported in recent studies[18,25,48–50]. The substrate, MWCNT, did not show any HER catalytic activity, while Ru@MWCNT exhibited excellent HER performance due to the presence of the small Ru nanoparticles (average 3.4 nm) stably anchored on its surface (Supplementary Fig. 7).

To evaluate the electrochemical surface area (ECSA) of the catalysts, the underpotential deposition of copper (Cu-UPD) on Ru@MWCNT and Pt/C were carried out. The ECSA of Ru@MWCNT was 7996.15 m$^2$g$^{-1}$$_{Ru}$, which was approximately two times higher than commercial Pt/C (3638.67 m$^2$g$^{-1}$$_{Pt}$) (Supplementary Fig. 8).

In order to identify the active sites on the Ru@MWCNT, thiocyanate ions ($^-$SCN), an active site toxin of metal catalysts, was added to the 0.5 M aq. H$_2$SO$_4$ electrolyte. The addition of $^-$SCN dramatically reduced the activity of the Ru@MWCNT, indicating that the Ru nanoparticles on the Ru@MWCNT were the active sites for HER catalysis (Supplementary Fig. 9).

To evaluate the long-term stability of Ru@MWCNT and Pt/C catalysts in both 0.5 M aq. H$_2$SO$_4$ (Figs. 4a) and 1.0 M aq. KOH solutions (Fig. 4b), cyclic stability tests were conducted at a scan rate of 100 mV s$^{-1}$. In acidic conditions, the commercial Pt/C showed an 8 mV negative shift at a current density of 10 mA cm$^{-2}$, while the Ru@MWCNT catalyst displayed only a 4 mV negative shift after 10,000 cycles. In alkaline conditions, the Ru@MWCNT exhibited 20 times better electrochemical stability than Pt/C (Fig. 4c). Stability was also examined via chronoamperometry technique at the applied potentials 20 and 35 mV, respectively, in acidic and alkaline media for 50 h, and the Ru@MWCNT exhibited no apparent loss in current density compared to Pt/C (Supplementary Fig. 10). In addition, TEM images of the Ru@MWCNT after the long-term stability test showed no change in morphology (Supplementary Fig. 11). These results indicate the exceptional stability of Ru@MWCNT compared to commercial Pt/C in both acidic and alkaline media.

For a fair comparison of catalytic activity, the polarization curves of Ru@MWCNT and Pt/C were normalized by ECSA. In 0.5 M aq. H$_2$SO$_4$ solution, the Ru@MWCNT showed slightly higher specific activity than Pt/C for a series of overpotentials (Fig. 4d). A more dramatic difference was observed in the specific activity between Ru@MWCNT (0.315 mA cm$^{-2}$) and Pt/C (0.122 mA cm$^{-2}$) in 1.0 M aq. KOH solution (Fig. 4d). At an overpotential of 30 mV, the Ru@MWCNT showed ~2.5 times higher specific activity than the Pt/C. This result indicates superior inherent catalytic activity, which is associated with the stronger H$_2$O binding energy and faster H$_2$O dissociation at the surface of the Ru nanoparticles on the Ru@MWCNT catalyst[25]. As a result, Ru@MWCNT can supply protons faster for more efficient hydrogen generation. Given its fast proton adsorption and reduction via appropriate hydrogen bond energy, fast proton supply, and rapid release of product (H$_2$), the Ru@MWCNT is a highly active HER catalyst.

To compare and evaluate the HER performance of the catalyst, we evaluated its turnover frequency (TOF), which is an important criterion for HER catalysts. TOF is the basis for determining inherent electrocatalytic efficiency, and the overpotential at 10 mA cm$^{-2}$ predicts the actual HER applicability. The TOF values for the active sites of the catalysts were calculated under acidic and alkaline conditions, following the previously reported method[22,23]. In 0.5 M aq. H$_2$SO$_4$ solution, the TOF value of Ru@MWCNT at 25 mV was 0.70 H$_2$ s$^{-1}$, which is very competitive compared to Pt/C (0.67 H$_2$ s$^{-1}$ at 25 mV) and other reported HER catalysts (Fig. 4e and Supplementary Table 4). In addition, in an alkaline solution, the TOF value of Ru@MWCNT at 25 mV was 0.40 H$_2$ s$^{-1}$, which is higher than that of Pt/C (0.25 H$_2$ s$^{-1}$) (Fig. 4e and Supplementary Table 5). The TOF values of the reference Pt/C are also reliable compared to other HER catalysts reported in recent studies (Supplementary Tables 4, 5). Hence, the TOF values in both conditions indicate that Ru@MWCNT outperforms Pt/C HER activity.

To further examine the catalysts from different perspectives, the mass activity of each catalyst was evaluated by normalizing the polarization curves with the masses of Ru and Pt. Mass activity is closely related to cost for practical applications. As shown in Fig. 4f, at the overpotential of 20 mV, the mass activity of Ru@MWCNT was 380 mA mg$^{-1}$$_{Ru}$ in acidic medium and 186 mA mg$^{-1}$$_{Ru}$ in alkaline medium. These values were much higher than Pt/C (165 and 52 mA mg$^{-1}$$_{Pt}$, respectively). Therefore, it can be safely stated that Ru@MWCNT has significant advantages over Pt/C in terms of overall catalytic performance and cost.

**The full water-splitting system analysis**. To further demonstrate an advanced practical use of Ru@MWCNT for water-splitting using an alkaline electrolyte (1.0 M aq. KOH solution), two-electrode devices with oxygen and hydrogen evolution electrodes were fabricated. The carbon papers (CPs) used as substrates for the electrodes were coated with the catalysts by electrospray (Supplementary Fig. 12). Both Ru@MWCNT and Pt/C coated on CP, as well as bare CP, was tested as a HER electrode. As the oxygen evolution reaction (OER) electrode, commercial iridium oxide (IrO$_2$) was coated on the CP. In order to accurately determine the actual amount of hydrogen generation, a systematic experiment was conducted by connecting a closed water-splitting device (HER + OER) directly to a gas chromatography (GC) instrument (Fig. 5a). The area of each electrode was 1 cm$^2$. Prior to the two-electrode evaluation, a three-electrode experiment was conducted to confirm the HER performance of the prepared electrodes. Current densities of devices with different HER electrodes were obtained (Fig. 5b). The Ru@MWCNT

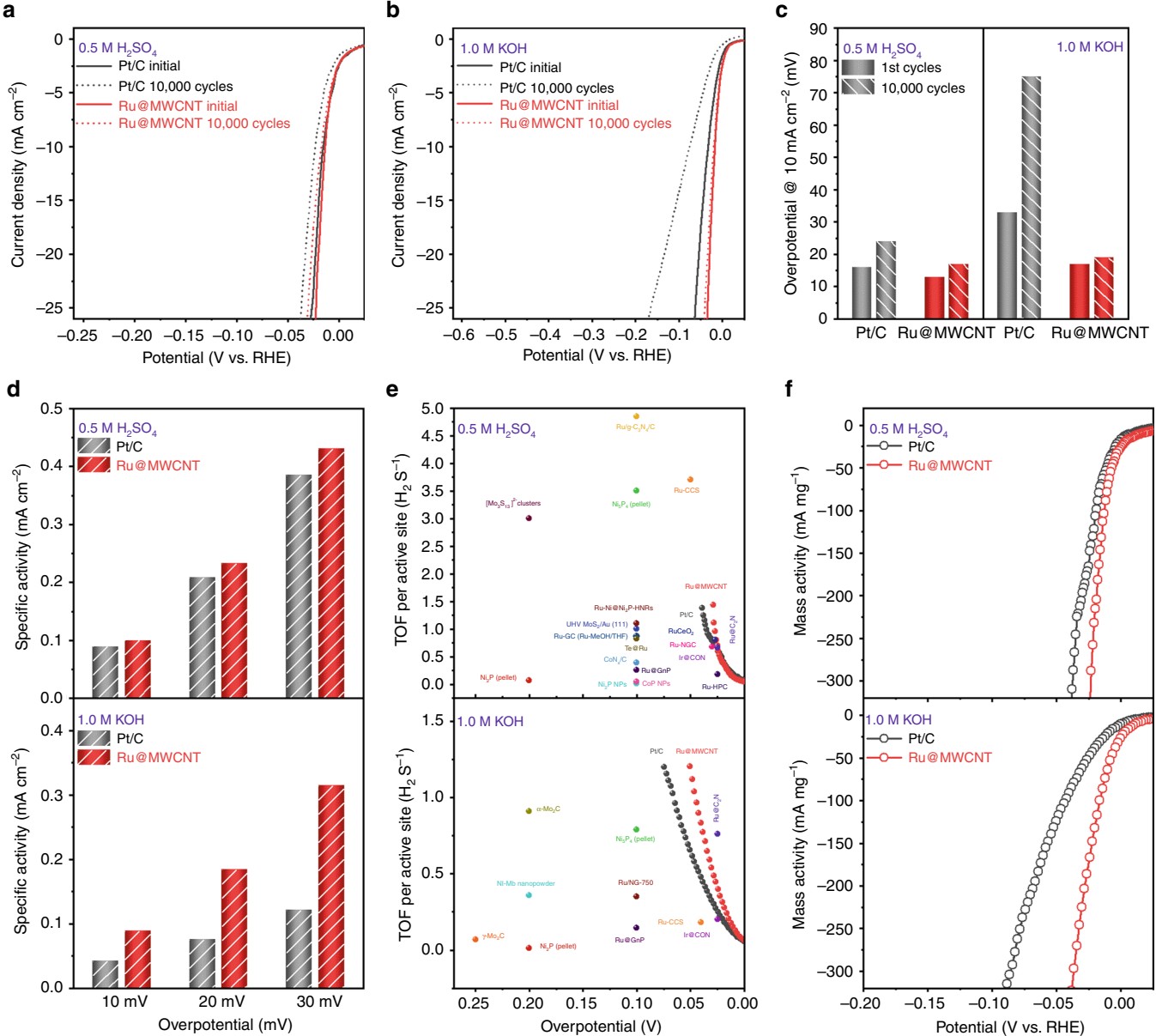

**Fig. 4 Comparison of electrochemical HER parameters. a**, **b** The polarization curves were recorded before and after 10,000 CV potential cycles. **c** Comparison of overpotential changes at 10 mA cm$^{-2}$ after 10,000 CV potential cycles in acidic and alkaline conditions. **d** Specific activities at different overpotentials (10, 20, and 30 mV) in N$_2$-saturated acidic and alkaline conditions, respectively. **e** Comparison of TOF values of the Ru@MWCNT and Pt/C with other recently reported HER catalysts in acidic and alkaline conditions, respectively. **f** Mass activities in N$_2$-saturated 0.5 M aq. H$_2$SO$_4$ and 1.0 M aq. KOH solutions, respectively.

electrode showed overpotentials of 10.4, 19.4, and 28.4 at 10, 20, and 30 mA cm$^{-2}$, respectively, while the Pt/C electrode showed 26.4, 40.4, and 50.4 mV at each corresponding current density.

A constant current was applied to the system for 20 h and the amount of hydrogen generated was measured every hour. As shown in Fig. 5c–e, the hydrogen production of the Ru@MWCNT per voltage was 2222.3, 3221.9, and 4194.0 μmol V$^{-1}$ meaning it produced 15.6% more than the Pt/C (Supplementary Tables 6–8). In addition, the hydrogen production of Ru@MWCNT per power consumption was also 15.4% higher than the Pt/C (Fig. 5f and Supplementary Table 9). Faradaic efficiency was also determined in the range of 1.5–1.8 V (Fig. 5g). The bare CP showed a Faradaic efficiency of only 11.4% at 1.8 V and there was no HER activity in the range of 1.5–1.7 V. The Pt/C electrode showed Faradaic efficiencies of 46.99, 81.98, 85.88, and 85.97% at 1.5, 1.6, 1.7, and

1.8 V, respectively, while the Ru@MWCNT electrode showed 85.88, 87.31, 92.24, and 92.28% at each corresponding voltage. Once again, the results indicated that the Ru@MWCNT catalyst was superior to the benchmark Pt/C (Supplementary Table 9).

The Ru@MWCNT catalyst was also coated on a large size titanium (Ti) mesh type electrode to check the practical application of the catalyst (Supplementary Video 1).

**Active site identification by DFT calculations.** First-principle density functional theory (DFT) calculations were also performed to gain more insight into the enhanced electrocatalytic activity of Ru@MWCNT active sites for hydrogen evolution reaction. It is widely known that the formation energy of metal-hydrogen (M–H) bond plays an important role in hydrogen evolution. Being at the center of volcano plot for electrocatalysts, Pt displays

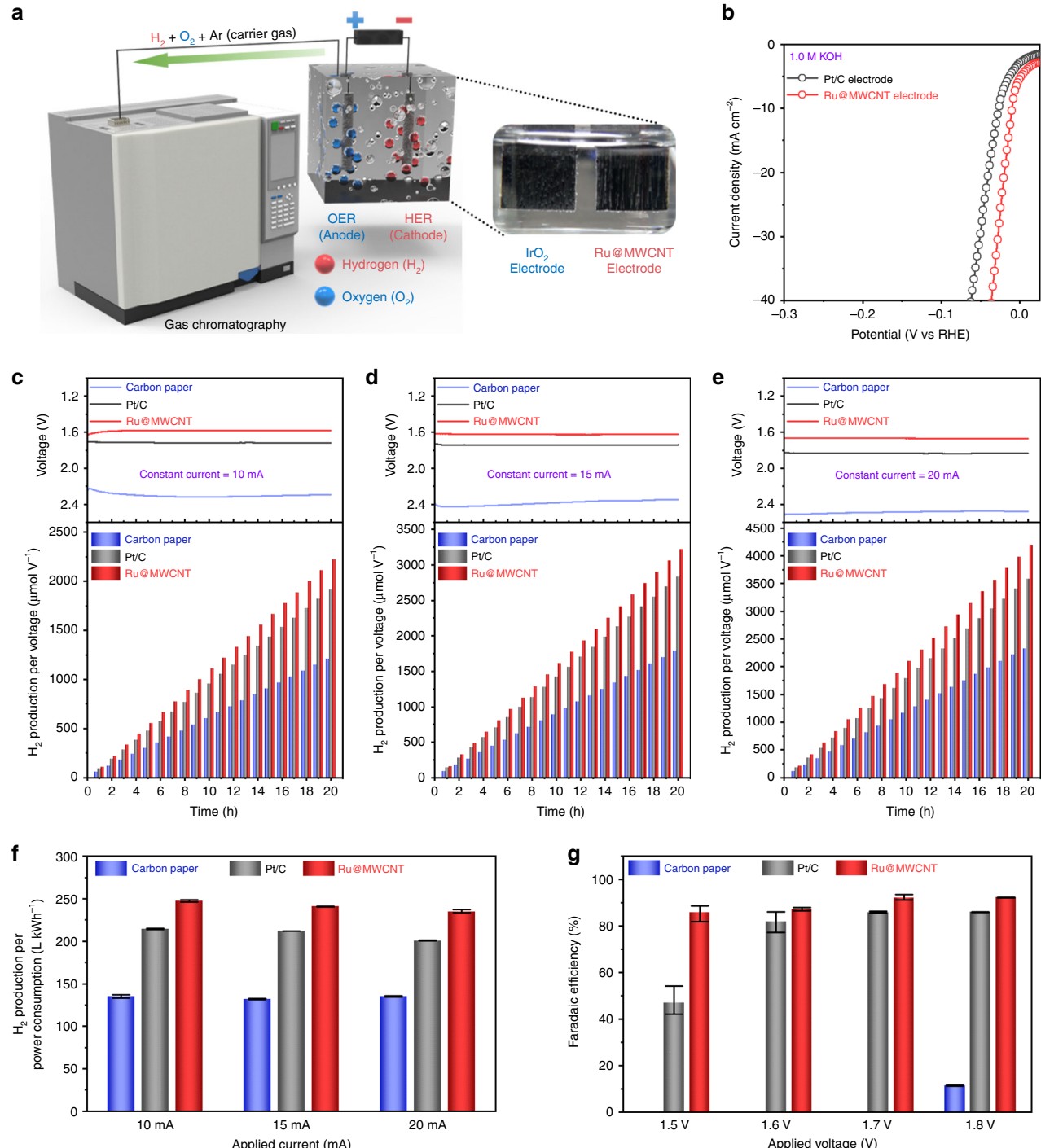

**Fig. 5 HER performance evaluation in actual water splitting. a** Schematic diagram of the two-electrode system measurement. **b** Polarization curves in $N_2$-saturated 1.0 M aq. KOH solution. Scan rate: 5 mV s$^{-1}$. **c–e** The voltage changes at constant current and the corresponding hydrogen production per voltage at specific currents of 10, 15, and 20 mA. **f** Hydrogen production per power consumption at specific current (10, 15, and 20 mA). The error bar reflects the three device results. **g** Faradaic efficiency at a specific voltage (1.5, 1.6, 1.7, and 1.8 V). The error bar reflects the three device results.

the optimal M−H binding energy, which is neither too weak nor too strong[25]. Catalysts having M−H binding energy similar or close to Pt−H (0.53 eV) will efficiently promote hydrogen evolution. The DFT calculations were performed based on previously reported Ru@C$_2$N for a clear comparison[25]. To sustain the catalytic activity of Ru nanoparticles, the important point is to prevent their aggregation (Ostwald ripening). The calculation showed that Ru nanoparticles on Ru@MWCNT have closer Pt

−H binding energy than on Ru@C$_2$N (Supplementary Fig. 14). This result indicates that Ru@MWCNT can have better HER performance than Ru@C$_2$N. For more details, hydrogen binding energies of possible H binding sites are identified, and the four most stable energies are 0.58, 0.64, 0.64, and 0.62 eV (Supplementary Fig. 15). All stable configurations of Ru@MWCNT show lower energies than Ru@C$_2$N (0.68 eV), suggesting that Ru@MWCNT can display enhanced catalytic activity. An

important point to be noted is that the Ru@MWCNT has an energy of −5.23 eV (10 Ru−C bonds) (Supplementary Fig. 16), implying that there are strong Ru–C bonds between Ru nanoparticles and MWCNT. This result reflects the stability of Ru nanoparticles on the surface of MWCNT (Ru@MWCNT) during long cycling test. Furthermore, the formation of Ru–C bonds was confirmed by EXAFS results (Supplementary Fig. 1), supporting that the aggregation (Ostwald ripening) of Ru nanoparticles can be hampered by forming strong bonds between Ru and MWCNT.

## Discussion

In summary, we have developed an efficient and stable HER catalyst for both acidic and alkaline media via a simple synthesis route. The catalyst consists of ruthenium (Ru) nanoparticles uniformly distributed and anchored on the surface of multiwalled carbon nanotubes (MWCNTs), or (Ru@MWCNT). The catalyst was realized by the formation of ruthenium carboxylate complexes, created between Ru ions ($Ru^{3+}$) and MWCNT-COOH, which was produced by the partial oxidation of MWCNT in nitric acid. Subsequent chemical ($NaBH_4$) and thermal reductions (annealing) turned the $Ru^{3+}$ ions into $Ru^0$ nanoparticles on the surface of MWCNT. The smaller particle size distribution and particle uniformity supports higher mass activity. MWCNT, which is widely known as a conductive material, not only provided an efficient catalytic support, stably anchoring the Ru nanoparticle active sites, but also fast electron transport.

As a result, the overall HER performance of the Ru@MWCNT, in terms of overpotential at 10 mA cm$^{-2}$, Tafel slope, and long-term stability, was superior to commercial Pt/C in both acidic and alkaline media. Regarding its potential value in practical applications, the Ru@MWCNT also displayed higher mass activity than commercial Pt/C. Most importantly, Ru@MWCNT has strong potential for mass production at low-cost, making it advantageous for use in practical applications. Last but not least, in an actual water-splitting experiment, Ru@MWCNT demonstrated an average Faradaic efficiency of 92.28% at 1.8 V, resulting in 15.4% higher hydrogen production per power consumption than Pt/C.

## Method

**Oxidation of multiwalled carbon nanotubes (MWCNT)**[51-54]. In a three-neck round bottom flask, MWCNT (10 g, CM-95, Hanwha Nanotech Co.) was dispersed in concentrated nitric acid (500 mL) after sonication for 1 h. Then, the reaction flask was placed in an oil bath and heated under reflux for 24 h. After cooling down to room temperature, the reaction mixture was poured into deionized water (1 L) and the precipitates, oxidized MWCNT, were collected by suction filtration. The product was further Soxhlet extracted with water and methanol to completely remove residual acid and other impurities, if any. The sample was finally freeze-dried for 3 days at –120 °C under reduced pressure.

**Preparation of Ru@MWCNT**. Partially oxidized MWCNTs (MWCNT-COOH, 10.0 g) and ruthenium chloride ($RuCl_3$, 3.0 g) were dispersed in N-methyl-2-pyrrolidone (NMP, 1.3 L). The mixture was agitated in a sonication bath for 3 h and further stirred overnight using a magnetic stirrer. The dispersed mixture was further sonicated for 2 h. Then, sodium borohydride (10% solution in NMP, 60 mL) was added using a dropping funnel under vigorous stirring. The mixture was stirred for 1 h and then mixed with acetone (1.5 L). The precipitates were collected by suction filtration and washed with water and freeze dried at –120 °C under reduced pressure for 3 days. The sample was annealed at different temperatures (600, 700, and 800 °C) under argon atmosphere for 2 h each. After annealing, the samples were further washed with water to remove unbound metal impurities in the matrix, if any. Finally, the samples were dried under reduced pressure.

**Electrochemical characterizations**. The electrochemical studies were carried out on an electrochemical workstation (Ivium, Netherlands) with a typical three-electrode cell. A graphite rod and an Ag/AgCl (saturated KCl) electrode were used as the counter electrode and reference electrode, respectively. All potentials were referenced with a reversible hydrogen electrode (RHE). Each catalyst (5 mg) was dispersed with Nafion (20 μL, 5 wt% in a mixture of lower aliphatic alcohol and water, Aldrich Chemical Inc.) in isopropyl alcohol (1.0 mL). The mixture was

sonicated for 30 min in an ice bath to form a uniform catalyst ink. The ink was drop casted onto a rotating ring-disk electrode (4 mm in diameter, RRDE) to form a film for the electrochemical tests. The loading amounts of each catalyst were 0.70 and 0.16 mg cm$^{-2}$ for the acidic and alkaline media, respectively. Linear sweep voltammetry (LSV) was conducted in both 0.5 M aq. $H_2SO_4$ and 1.0 M aq. KOH solutions at a scan rate of 5 mV s$^{-1}$. The solution resistances ($R_s$) in the 0.5 M aq. $H_2SO_4$ and 1.0 M aq. KOH solutions were 15 and 17 Ω, respectively, tested by Nyquist plots. All data were further used for the Ohmic drop (iR) correction. The reference electrode was calibrated, and all potentials were referenced to a RHE (Supplementary Fig. 13).

**Active sites calculations**. The underpotential deposition (UPD) of copper (Cu) was used to calculate the active sites of the Ru@MWCNT and Pt/C. In this method, the number of active sites (n) can be calculated based on the UPD copper stripping charge ($Q_{Cu}$, $Cu_{upd} \rightarrow Cu^{2+} + 2e^-$) using the following equation[25].

$$n = Q_{Cu}/2 F$$

where F is the Faraday constant (96,485.3 C mol$^{-1}$).

**Measurement of the turnover frequency (TOF)**. The TOF (s$^{-1}$) was calculated with the following equation.

$$TOF = I/(2Fn)$$

where I is the current (A) during linear sweep voltammetry (LSV), F is the Faraday constant (96485.3 C mol$^{-1}$), n is the number of active sites (mol). The factor 1/2 is based on the assumption that two electrons are necessary to form on hydrogen molecules.

Ru@MWCNT was first polarized at 0.22, 0.23, 0.24, 0.25, 0.26, 0.27, 0.28, 0.29, 0.30, and 0.31 V for 100 s (Supplementary Fig. 8a). For the given polarization potential, there were only two oxidation peaks related to bulk and monolayer of Cu. To obtain monolayer of copper, 0.26 V was selected in the following test for Ru@MWCNT (Supplementary Fig. 8b, c).

**Preparation of HER electrodes**. Each catalyst (5 mg, Ru@MWCNT, Pt/C or $IrO_2$) was dispersed in isopropyl alcohol (1.0 mL) after applying sonication for 30 min. The resultant catalyst ink was directly deposited onto carbon paper (CP) using an electrospray method. First, each catalyst ink was placed into a plastic syringe equipped with a 30-gauge stainless steel hypodermic needle. The needle was connected to a high voltage power supply (ESN-HV30). A voltage of ~4.3 kV was applied between a metal orifice and the conducting substrate at a distance of 8 cm. The feed rate was controlled by the syringe pump (KD Scientific Model 220) at a constant flow rate of 20 μL min$^{-1}$. The electric field overcomes the surface tension of the droplets, resulting in the minimization of numerous charged mists. Each electrode was tested after drying in vacuum oven at room temperature for 1 day.

**Computation method**. To simulate the experimental results, an icosahedral symmetric $Ru_{13}$ nanoparticle is attached to wall of carbon nanotube (CNT). The Vienna Ab initio Simulation Package (VASP) calculations are carried out to obtain the ground state of many electrons system in the framework of density functional theory[55-57]. The plane-wave basis set with an energy cutoff of 500 eV and the PBE-type gradient-corrected exchange-correlation potential suggested by Perdew, Burke, and Ernzerhof were employed[58]. The ionic potentials were described by the projector-augmented wave potentials, and the atomic configurations were selectively relaxed with the residual forces smaller than 0.001 eV/Å[56]. Periodic boundary conditions for DWCNT are made by 20 Å × 25 Å × 14.85 Å unit cell and 1 × 1 × 6 k-points sampling. In order to reduce the calculation cost, we cut the fully optimized DWCNT into half and get a semi-cylinder shape with 108 carbon atoms. The carbons further from $Ru_{13}$ nanoparticle bonding region are fixed and all the other atoms are relaxed in geometric optimization.

**Materials characterizations**. The morphologies of the samples were studied by FE-SEM (Nanonova 230, FEI, USA) and high-resolution transmission electron microscopy (HR-TEM, JEM-2100F, JEOL, Japan). Specific surface area was determined by nitrogen adsorption–desorption isotherms, using the BET method (BELSORP-max, BEL, Japan). Thermogravimetric analysis (TGA) was performed at a ramping rate of 10 °C min$^{-1}$ in air on a thermogravimetric analyzer (Q200, TA, USA). X-ray diffraction (XRD) patterns were recorded on a high-power X-ray diffractometer (D/MAZX 2500 V/PC, Rigaku, Japan), using Cu-Kα radiation (35 kV, 20 mA, λ = 1.5418 Å). An X-ray photoelectron spectrometer (XPS, K-alpha, Thermo Fisher Scientific, UK) and elemental analysis (EA, Flash 2000 Analyzer) were employed to determine chemical composition. The electrochemical HER test was initiated, and the evolved hydrogen gas was analyzed by gas chromatography (GC-2010 Plus, Shimadzu, Japan), with a thermal conductivity detector (TCD). Argon was used as the carrier gas. X-ray absorption fine spectra of the prepared catalysts were collected in the transmission mode using ionization detectors (Oxford) at the Pohang Accelerator Laboratory (PAL). The X-ray absorption spectra for the Ru K edge were acquired at room temperature using beamline 6D of PAL, where their X-ray energies from the EXAFS analysis were calibrated with Ru foil. Background subtraction, normalization and Fourier transformation (FT) were done by standard procedures with ATHENA program. The extracted EXAFS signal, χ(r) and k$^2$χ(k) were analyzed for all samples. The selected k ranges for Ru

acetylacetonate, Ru@MWCNT (before heat-treatment) and Ru@MWCNT (after heat-treatment) in plotting the Ru K-edge graphs were 3.0–11.0, 3.0–11.0, and 3.0–8.7, respectively, and the selected R range is 1.0–3.0 for all samples.

## Data availability

The data supporting this study are available from the corresponding author upon reasonable request.

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

## Acknowledgements

This research was supported by the Creative Research Initiative (CRI, 2014R1A2069102), BK21 Plus (10Z20130011057), Science Research Center (SRC, 2016R1A5A1009405) and Young Researcher (2019R1C1C1006650) Programs through the National Research Foundation (NRF) of Korea, funded by the Ministry of Science, ICT, and Future Planning. M.S.O. and N.P. appreciate NRF for the support (2019R1A2C2089332). The EXAFS experiments were performed in the PAL beamline (6D C&S UNIST-PAL).

## Author contributions

J.-B.B, J.M., and D.H.K. conceived and designed the project. D.H.K, J.M., J.-P.J., and H.-J.N carried out the synthesis and characterizations. S.-J,K and D.H.K were involved in the EXAFS characterizations. D.H.K performed electrochemical analyses. M.S.O. and N.P. carried out the DFT study. D.H.K, J.M., and J.-B.B. wrote the manuscript and all authors discussed the results and commented on the paper.

## Competing interests

The authors declare no competing interests.
