## [Peer Review File · Nature Communications]

Reviewers' comments:

Reviewer #1 (Remarks to the Author):

The manuscript by Mahmood, Baek and co-workers deals with the development of stable and highly active HER cathodes based on Ru NPs onto multiwalled carbon-nanotubes and their inclusion in an overall water splitting device for the generation of H₂ in alkaline media at high Faradaic efficiency. The authors claim that the reported Ru@MWCNTs system shows excellent performance (in terms of overpotentials, specific activities and TOF values) in HER at both acidic and alkaline conditions, outperforming commercial Pt/C. This is professionally and convincingly demonstrated in the manuscript by benchmarking the reported system (and the reference Pt/C) by well-established electrochemical methods. The care with that the authors tackle this benchmarking by tracking Ru active sites and ECSA by Cu-UPD methods is remarkable, not common for supported systems like the one reported. The interest of Ru-based systems for HER, particular in alkaline conditions where they show by far more stable than Pt-based systems, is not new and has been recently reviewed (see for instance ChemSusChem 2019, 12, 2493-2514 / ACS Catal. 2019, DOI: 10.1021/acscatal.9b02457). The preparation of Ru NPs stabilized onto carbon-based supports from the reductive decomposition of pre-arranged RuCl₃ has also been reported by similar synthetic methods (see for instance: Angew. Chem. Int. Ed. 2018, 57, 5848). Thus, the manuscript does not contribute with new concepts to the research field from these points of view. However, additionally to lower overpotential values, the reported Ru@MWCNTs system outperform Pt/C in terms of intrinsic activity of the active sites in both acidic and alkaline media (TOFs, exchange current density). This unusual fact, together with the practical inclusion of the reported cathodes in overall water splitting devices of almost quantitative Faradaic efficiency and very high H₂ production per consumed power (superior to that of the reference Pt/C anode), makes the manuscript susceptible to be published in Nature Communications after considering the following points:

1) The authors should clarify if the surface oxidation state of the Ru-based NPs evolve with time when exposed to air. Is the XPS analysis provided in Figure 3 of the Supplementary Information obtained after air-exposition of the hybrid system? The 3d_{5/2} peak at 280.4 eV observed (Figure 3, supplementary info) could in fact correspond to a mixed Ru/RuO₂ phase (see BE for Ru and RuO₂ in Surf. Interface Anal. 2015, 47, 1072-1079). Did the authors check the deconvolution of this peak by fitting a mixture of Ru and RuO₂? Echem analysis is carried out under N₂ atmosphere. Does this have anything to do with the potential evolution of the Ru metal NPs under air conditions? The Ru/RuO₂ surface ratio and its evolution with time under air and under turnover conditions is key in Ru-based systems for HER and should be carefully analysed.

2) The comparison of overpotentials (Figure 3, g-h) and TOFs (Figure 4e) with related/representative reported systems is incomplete. In terms of overpotentials, the following relevant references should be included: Nano Energy 2019, 58, 1-10 & ACS Catal. 2018, 8, 11094-11102). Given key role of Ru-based systems in HER catalysis, for a fair comparison of the here reported system a set of relevant Ru-based electrocatalysts studied both under acidic and alkaline conditions should be included in the TOF analysis together with the data for Pt/C and 1st-row based electrocatalysts. As exposed above, these data have been recently reviewed in ChemSusChem 2019, 12, 2493-2514 / ACS Catal. 2019, DOI: 10.1021/acscatal.9b02457. Particularly, the following systems show TOF data competitive with the values reported by the authors and merit inclusion in Figure 4e:

- Acidic conditions: ACS Catal. 2018, 8, 5714 / J. Am. Chem. Soc. 2018, 140, 2731 / ACS Catal. 2018, 8, 11094 / Adv. Mater. 2018, 40, 1800047 / J. Mat. Chem. A 2018, 6, 2311.

- Alkaline conditions: Nano Energy 2019, 58, 1-10 / Adv. Mater. 2018, 40, 1800047 / Angew. Chem. Int. Ed. 2018, 57, 5848 / ACS Appl. Mater. Interfaces 2017, 9, 3785.

Minor points:

1) Repetition should be avoided between Figure captions and manuscript text. An example is found for instance in the caption of Figure 1.

- 2) Page 2, second paragraph: the media where Pt shows poor electrochemical stability (alkaline conditions) should be detailed.
- 3) Transferring the numeric scale bars from Figure captions to TEM images would ease the reading of the manuscript.
- 4) The stability analysis through repetitive LSV cycle in acidic conditions should be added to the Supplementary Information (Figure S10).
- 5) The applied potential value should be noted in Figure S10 as well as in the manuscript text description.

Reviewer #2 (Remarks to the Author):

Comments:

This paper reports on an efficient but simple method of synthesizing a HER catalyst, consisting of ruthenium (Ru) nanoparticles uniformly deposited on the surface of multi-wall carbon nanotubes (MWCNTs). This catalyst shows excellent performance for HER. I would like to recommend its acceptance for publication in this journal after the following major revisions:

1. The Ru content should be provided by ICP measurement.
2. The particle size distribution of the nanomaterials should be provided in Figure 2c.
3. The Ru-based electrocatalysts for HER were reported in previous literatures. In particular, what are the difference and innovation of this paper compared to those (Nat. Nanotech. 2017, 12, 441; Chem. Eur. J. 2019, 25, 8579-8584, Nat. Commun. 2019, 10, 631)?
4. There are many minor mistakes in the References, I just listed some as below but suggest the authors to check throughout the References and may find more:

1). Page 15, ACS Catalysis 4, 1693-1705 (2014).

Comment: please replace "ACS Catalysis" by "ACS Catal."

2). Page 15, the Hydrogen Evolution Reaction at All Ph Values. Angew. Chem. Int. Ed. 56, 11559-11564 (2017).

Comment: please replace "All Ph Values" by "All pH Values"

3) Page 15, Mahmood J. et al. An Efficient and Ph-Universal Ruthenium-Based Catalyst for the Hydrogen Evolution Reaction. Nat. Nanotechnol. 12, 441 (2017).

Comment: please replace "Ph-Universal" by "pH-Universal".

Reviewer #3 (Remarks to the Author):

In this paper, Kweon and his co-workers have reported their work entitled "Ruthenium@carbon nanotube electrocatalyst with superior Faradaic efficiency for hydrogen production". The hybrid sample of ultrathin Ru nanoparticles on MWCNT (Ru@MWCNT) was prepared through a simple reduction method with the aid of sodium borohydride (NaBH₄). Fundamental morphology, structure, catalytic performance and stability of the material in both alkaline and acidic media were studied. Ru@MWCNT displays outstanding HER performance in both acidic and alkaline solutions in almost all aspects such as the overpotential at a current density of 10 mA cm⁻², Tafel slope, the exchange current density, charge transfer resistance, ECSA, and stability, TOF, Faradaic efficiency, as well as various specific activities, etc., obviously superior to benchmark Pt/C. Although the author provides a relatively complete study on the Ru@MWCNT materials for hydrogen evolution in both acidic and alkaline solution, the paper does not provide sufficiently the technical-novelty or scientific achievement in the development of Ru-based HER catalysts. First, the preparation method by the reduction method with the help of NaBH₄ has been widely used in previous works, for instance Adv. Energy Mater. 2018, 1801698, Nature nanotechnology 12.5 (2017): 441. Second, only some basic characterization techniques were used which cannot provide insights in the materials. The author employed

thiocyanate ions ($-\text{SCN}$) to identify the active sites on the Ru@MWCNT. In order to provide the in-depth understanding about the catalyst for HER, the authors should pay more attention to more advanced characterization techniques such as Operando XPS or X-ray absorption spectroscopy (XAFS) or DFT calculations to reveal the real active sites. Lastly but most importantly, recently, there are many similar literatures about the Ru-C-based materials as HER catalysts in pH-universal electrolytes (Energy Environ. Sci., 2018, 11, 800; Adv. Energy Mater. 2018, 1801698; J. Mater. Chem. A, 2018, 6, 2311; Adv. Mater. 2018, 30, 1803676; Sustainable Energy Fuels, 2017, 1, 1028; J. Mater. Chem. A, 2017, 5, 25314; Chem. Commun., 2018, 54, 13076; Catal. Sci. Technol., 2017, 7, 4964 etc.). Considering the insufficient novelty of this study, the manuscript was not suitable to be published in Nature Communications.

Reviewer #4 (Remarks to the Author):

In the manuscript, the authors claimed a Ruthenium@carbon nanotube electrocatalyst for HER with superior faradaic efficiency. Yet, there are several points that the authors should be noticed clearly.

1. So many Ru based catalysts for HER have been reported. This work doesn't demonstrate their novelty in the material design.
2. As for the synthetic route, this work doesn't show its advantages compared to other noble-metal based catalysts which shown in Supp. Table S2,3.
3. The electrochemical investigation could not satisfy the performance explanation, further theoretical explanation should be involved.

In summary, I think the novelty and quality of this work do not reach the level of Nature Commun.

Reviewers' comments:

Reviewer #1 (Remarks to the Author):

Comment 1.0. The manuscript by Mahmood, Baek and co-workers deals with the development of stable and highly active HER cathodes based on Ru NPs onto multiwalled carbon-nanotubes and their inclusion in an overall water splitting device for the generation of H₂ in alkaline media at high Faradaic efficiency. The authors claim that the reported Ru@MWCNTs system shows excellent performance (in terms of overpotentials, specific activities and TOF values) in HER at both acidic and alkaline conditions, outperforming commercial Pt/C. This is professionally and convincingly demonstrated in the manuscript by benchmarking the reported system (and the reference Pt/C) by well-established electrochemical methods. The care with that the authors tackle this benchmarking by tracking Ru active sites and ECSA by Cu-UPD methods is remarkable, not common for supported systems like the one reported. The interest of Ru-based systems for HER, particular in alkaline conditions where they show by far more stable than Pt-based systems, is not new and has been recently reviewed (see for instance ChemSusChem 2019, 12, 2493-2514 / ACS Catal. 2019, DOI: 10.1021/acscatal.9b02457). The preparation of Ru NPs stabilized onto carbon-based supports from the reductive decomposition of pre-arranged RuCl₃ has also been reported by similar synthetic methods (see for instance: Angew. Chem. Int. Ed. 2018, 57, 5848). Thus, the manuscript does not contribute with new concepts to the research field from these points of view. However, additionally to lower overpotential values, the reported Ru@MWCNTs system outperform Pt/C in terms of intrinsic activity of the active sites in both acidic and alkaline media (TOFs, exchange current density). This unusual fact, together with the practical inclusion of the reported cathodes in overall water splitting devices of almost quantitative Faradaic efficiency and very high H₂ production per consumed power (superior to that of the reference Pt/C anode), makes the manuscript susceptible to be published in Nature Communications after considering the following points:

Response 1.0. We appreciate the reviewer #1 for his/her thoughtful comments and critical assessment. The reviewer's suggestions and criticisms help us to substantially improve the quality of manuscript. As the reviewer pointed out, the researches on Ru-based catalysts have been very active in recent years and have made significant progress (*Nat. Commun.* 2019, 10, 631; *Adv. Mater.* 2018, 30, 1800047; *Energy Environ. Sci.* 2018, 11, 1232–1239; *Nat. Nanotechnol.* 2017, 12, 441–446).

The originality of our current catalyst, Ru@MWCNT, is the simplicity and practicality in synthesis. The synthesis involves a mild oxidation of multi-walled carbon nanotube (MWCNT) to introduce carboxylic acids on its outmost surface. Carboxylic acid functionalized MWCNT (MWCNT–COOH) can efficiently coordinate with Ru precursor (Ru³⁺) by forming Ru carboxylate complexes. Subsequent chemical and thermal reduction of Ru³⁺ turned into zero-valent Ru (Ru⁰) nanoparticles, which can be stably anchored on the surface of MWCNT. Hence, the introduction of carboxylic acids plays an important role in forming smaller, more uniform and stable Ru nanoparticles on the surface of MWCNT (Ru@MWCNT). In this way, Ru@MWCNT can demonstrate better catalytic performance and stability than the state-of-art commercial Pt/C.

In addition, most of previous studies evaluate the activity of HER catalysts using half-cell system (three-electrode system consisting of working electrode, reference electrode and counter electrode). As mentioned above, the HER field is now approaching a commercial stage based on recent progress. There are three important criteria to meet commercial demands, which are cost competitiveness, performance and stability. We focused on developing a catalyst to satisfy these factors.

First, to be cost competitive, we need to reduce the content of Ru and the cost of supports. In this regard, we designed a relatively inexpensive catalyst for commercialization by reducing Ru loading and using MWCNT. MWCNT can serve as an efficient catalytic support with easy functionalization and good electrical conductivity.

Supplementary Fig. 1 | **a**, Chemical structure of ruthenium acetylacetonate. **b**, Fourier transformed (FT) Ru K-edge EXAFS spectra of before/after heat-treated Ru@MWCNT and reference Ru acetylacetonate.

Second, in order to enhance performance and stability, it is necessary to stably form small and uniform Ru nanoparticles on catalytic support (MWCNT). Chemistry of carboxylic acids (–COOH) and Ru³⁺ was applied for the stable formation of small and uniform Ru nanoparticles on the surface of MWCNT (Ru@MWCNT). To clarify this, EXAFS results were added to identify the formation of Ru carboxylate and local structural environment of Ru on Ru@MWCNT before and after heat-treatments (**Supplementary Fig. 1**). It was found that the Ru@MWCNT before heat-treatment has a very similar Ru coordination compared to the reference Ru acetylacetonate.

Third, we performed a first-principle density functional theory (DFT) calculations to further understand the active sites and origin of enhanced HER electrocatalytic activity of Ru@MWCNT with stability (**Supplementary Fig. 14, 15 and 16**). DFT calculations were closely compared with our previously reported Ru@C₂N as a reference (*Nat. Nanotechnol.*

2017, 12, 441-446). The results indicated that Ru@MWCNT has the more favorable for hydrogen adsorption, suggesting the better catalytic activity. In addition, the higher stability of Ru@MWCNT was also anticipated by forming the stronger interaction between Ru and MWCNT.

Supplementary Fig. 14 | Hydrogen adsorption configuration on the surfaces of model catalysts and their Ru-H binding energies: **a**, Ru₁₃, **b**, Ru₁₃@MWCNT, **c**, Ru₁₃@C₂N.

Supplementary Fig. 15 | Hydrogen adsorption configuration at different Ru-H bonding sites on the surface of Ru@MWNCT. **a**, 0.58 eV/H, **b**, 0.64 eV/H, **c**, 0.64 eV/H, **d**, 0.62 eV/H.

Supplementary Fig. 16 | Formation energy of Ru nanoparticles on the surface of MWCNT.

Ten Ru-C bonds are formed with the release of energy (5.23 eV).

Finally, Ru@MWCNT was evaluated by a full water splitting system (**Fig. 5a**). Like other energy-related fields, such as solar cells and fuel cells, catalysts themselves are evaluated using half-cell system. Then, full-cell performances are also determined (*Chem. Rev.* 2015, 115, 4823-4892; *Adv. Mater.* 2019, 31, 1804440). In this study, we analysed hydrogen production per power consumption and Faradaic efficiency, which are key parameters for commercialization. An evaluation of full water splitting system provides an insight for the further growth in the HER field in practice. However, most of the reported literatures fail to provide these kinds of information. With this in mind, we built a full water splitting system and fabricated the HER electrode using carbon paper (1 cm²). The hydrogen production per power consumption and Faradaic efficiency were evaluated using GC (**Fig. 5**). Furthermore, the HER electrode using titanium (Ti)-mesh (50 cm²) was also prepared and tested (**Supplementary Video 1**). Therefore, this work suggests a standard evaluation method to reinforce its quality.

Comment 1.1. The authors should clarify if the surface oxidation state of the Ru-based NPs evolve with time when exposed to air. Is the XPS analysis provided in Figure 3 of the Supplementary Information obtained after air-exposition of the hybrid system? The 3d5/2 peak at 280.4 eV observed (Figure 3, supplementary info) could in fact correspond to a mixed Ru/RuO₂ phase (see BE for Ru and RuO₂ in *Surf. Interface Anal.* 2015, 47, 1072-1079). Did the authors check the deconvolution of this peak by fitting a mixture of Ru and RuO₂? Echem analysis is carried out under N₂ atmosphere. Does this have anything to do with the potential evolution of the Ru metal NPs under air conditions? The Ru/RuO₂ surface ratio and its evolution with time under air and under turnover conditions is key in Ru-based systems for HER and should be carefully analysed.

Response 1.1. Many thanks to the specific technical comments. In the quoted papers, peaks for Ru (metallic Ru⁰) and RuO₂ (Ru⁴⁺) are at 279.75 and 281.37 eV, respectively. In our paper, Ru nanoparticles are anchored on MWCNT. As a result, Ru has a slight peak shift due to the

interaction between Ru and carbon. To identify the peak position of the Ru-carbon interaction, the XPS spectra were compared between commercial Ru/C and Ru@MWCNT (**Fig. R1**, *ACS Sustain. Chem. Eng.* 2018, 6, 4390-4399; *Ind. Crop. Prod.* 2017, 97, 10-20).

Fig. R1 | **a**, XPS spectra of Ru 3d_{5/2} for various Ru/C catalysts (*ACS Sustain. Chem. Eng.* 2018, 6, 4390-4399). **b**, XPS spectra of Ru 3d for Ru catalysts on different supports based on oxidation states: (left) Ru/C (J.M.); (right) Ru/C (Sigma). (*Ind. Crop. Prod.* 2017, 97, 10-20).

There are several reasons for the electrochemical analysis in an N₂ or argon (Ar) atmosphere. First, to keep the pH constant and to efficiently evacuate the hydrogen produced. Second, there are dissolved gases including oxygen (O₂) in aqueous electrolyte solution. Dissolved oxygen is involved in the oxygen reduction reaction (ORR). As shown in **Fig. R2** and **Table R1** below. An overlap between ORR and HER in the negative potential region is observed. For example, commercial Pt/C is active for both ORR and HER, indicating that the ORR can affect the HER (*Chem. Soc. Rev.* 2015, 44, 2060-2086). In general, N₂ or Ar is recommended in aqueous electrochemical systems to remove dissolved oxygen. In addition, N₂ does not contribute to the HER mechanism (below equation). Hence, experiments are conducted in an N₂ atmosphere to precisely assess HER performance. Many previously reported literatures on HER have also conducted electrochemical analysis in N₂ or Ar atmosphere for the reliable assessment of HER performance

Table R1 | Hydrogen evolution reaction and oxygen reduction reaction mechanism in acidic and alkaline electrolytes

Acidic electrolyte (Hydrogen Evolution Reaction)

Volmer step

Tafel step

Heyrovsky step

Alkaline electrolyte (Hydrogen Evolution Reaction)

Volmer step – water dissociation

Tafel step

Heyrovsky step

Acidic electrolyte (**Oxygen Reduction Reaction**)

Four-electron process

Two-electron process

Alkaline electrolyte (**Oxygen Reduction Reaction**)

Four-electron process

Two-electron process

Fig. R2 | The polarization curves for two pairs of the key energy-related electrochemical reactions and their overall reaction equations. Red and blue curves refer to the hydrogen-involving and oxygen-involving reactions, respectively. The lines are not drawn to scale (*Chem. Soc. Rev.* 2015, 44, 2060-2086).

Comment 1.2. The comparison of overpotentials (Figure 3, g-h) and TOFs (Figure 4e) with related/representative reported systems is incomplete. In terms of overpotentials, the following relevant references should be included: Nano Energy 2019, 58, 1-10 & ACS Catal. 2018, 8, 11094-11102). Given key role of Ru-based systems in HER catalysis, for a fair comparison of the here reported system a set of relevant Ru-based electrocatalysts studied both under acidic and alkaline conditions should be included in the TOF analysis together with the data for Pt/C and 1st-row based electrocatalysts. As exposed above, these data have been recently reviewed in ChemSusChem 2019, 12, 2493-2514 / ACS Catal. 2019, DOI: 10.1021/acscatal.9b02457. Particularly, the following systems show TOF data competitive with the values reported by the authors and merit inclusion in Figure 4e:

- Acidic conditions: ACS Catal. 2018, 8, 5714 / J. Am. Chem. Soc. 2018, 140, 2731 / ACS Catal. 2018, 8, 11094 / Adv. Mater. 2018, 40, 1800047 / J. Mat. Chem. A 2018, 6, 2311.
- Alkaline conditions: Nano Energy 2019, 58, 1-10 / Adv. Mater. 2018, 40, 1800047 / Angew. Chem. Int. Ed. 2018, 57, 5848 / ACS Appl. Mater. Interfaces 2017, 9, 3785.

Response 1.2. We agree with the reviewer's comment. Accordingly, we revised the overpotential (**Fig. 3d and Supplementary Table 2 and 3**) and TOF (**Fig. 4e and Supplementary Table 4 and 5**) values, including the data in the suggested papers for comparison with recently competitive catalysts.

Minor points:

Comment 1.3. Repetition should be avoided between Figure captions and manuscript text. An example is found for instance in the caption of Figure 1.

Response 1.3. We agree with the reviewer's comments and modified the repetitive content between the figure captions and the text in the revised manuscript.

Comment 1.4. Page 2, second paragraph: the media where Pt shows poor electrochemical stability (alkaline conditions) should be detailed.

Response 1.4. We added further details, why Pt shows poor stability and related references. The revised paragraph is below.

“However, in addition to soaring cost and scarcity, Pt has poor electrochemical stability, which is associated with leaching in corrosive electrolytes and irreversible aggregation of Pt nanoparticles by Ostwald ripening (*Energy Environ. Sci.* 2019, 12, 1000-1007; *ACS Catal.* 2015, 5, 4819-4824), limiting its practical applications.”

Comment 1.5. Transferring the numeric scale bars from Figure captions to TEM images would ease the reading of the manuscript.

Response 1.5. Following the reviewer's thoughtful suggestion, scale bars are added to TEM and STEM images for better readability.

Comment 1.6. The stability analysis through repetitive LSV cycle in acidic conditions should be added to the Supplementary Information (Figure S10).

Response 1.6. Stability analysis with repetitive LSV cycles in acidic conditions is shown in **Fig 4a**. If the comment is related to chronoamperometry (current-time) data, we have added stability analysis by chronoamperometry (current-time) method in acidic conditions in the Supplementary Information (**Supplementary Fig. 10**).

Comment 1.7. The applied potential value should be noted in Figure S10 as well as in the manuscript text description.

Response 1.7. Thanks for suggestion, the applied potential value is added in the revised text and **Supplementary Fig. 10**.

Reviewer #2 (Remarks to the Author):

Comments:

This paper reports on an efficient but simple method of synthesizing a HER catalyst, consisting of ruthenium (Ru) nanoparticles uniformly deposited on the surface of multi-wall carbon nanotubes (MWCNTs). This catalyst shows excellent performance for HER. I would like to recommend its acceptance for publication in this journal after the following major revisions:

Comment 2.1. The Ru content should be provided by ICP measurement.

Response 2.1. We are grateful to the reviewer #2 for sparing precious time to evaluate our manuscript. Based on the reviewer's constructive comments, we made necessary amendments in the revised manuscript. Although ICP is usually used to detect minute amount of metals in the material, TGA and EA are better to quantify relatively high metal contents in the material. In fact, during pre-treatment of ICP, aqua regia (hydrochloric acid and nitric acid mixture) or nitric acid is used as solvent. Ru is barely soluble in aqua regia and common acidic solvents (<https://en.wikipedia.org/wiki/Ruthenium>, DOI: 10.5772/intechopen.76393). ICP can only detect Ru, but it cannot quantify the high loading amount of Ru in this work. Therefore, we think that determining Ru content using ICP in this work is not reliable. Nevertheless, we provide the ICP results of Ru@MWCNT for the referee's consideration (**Fig. R3**).

In addition, unlike Pt (quite soluble in ICP solvents), the insolubility of Ru is, in turn, the stability of Ru-based electrocatalysts. Hence, Ru@MWCNT catalyst displays more stable than Pt/C in both corrosive acidic and alkaline electrolytes.

Ru 267.876 Calibration (mg/kg) 11/12/2019, 2:14:26 PM				Correlation Coefficient: 0.999982		
Label	Flags	Int. (c/s)	Std Conc.	Calc Conc.	Error	%Error
Blank		33.6	0.000	0.000	-	-
Standard 1		6044	1.00	1.01	0.006	0.6
Standard 2		30177	5.00	5.05	0.046	0.9
Standard 3		59633	10.0	9.98	-0.024	-0.2

Curve Type: Linear

Equation: $y = 5974.0 x + 33.6$ 
#1 (Samp)		11/12/2019, 2:17:52 PM			Tube 9	
Weight: 0.0203		Volume: 50			Dilution: 1	
Label	Sol'n Conc.	Units	SD	%RSD	Int. (c/s)	Calc Conc.
Ru 267.876	0.167	mg/kg	0.006	3.9	1029	411 mg/kg

#2 (Samp)		11/12/2019, 2:18:57 PM			Tube 10	
Weight: 0.0199		Volume: 50			Dilution: 1	
Label	Sol'n Conc.	Units	SD	%RSD	Int. (c/s)	Calc Conc.
Ru 267.876	0.221	mg/kg	0.009	4.0	1352	555 mg/kg

Fig. R3 | ICP analysis of Ru@MWCNT.

Comment 2.2. The particle size distribution of the nanomaterials should be provided in Figure 2c.

Response 2.2. The particle size distribution of Ru nanoparticle is already given the inset of **Fig. 2d**. **Fig. 2c** is related to nitrogen adsorption-desorption isotherm. We assume that the reviewer #2 is looking for the pore size distribution of Ru@MWCNT, which we have added an inset in **Fig. 2c**.

Comment 2.3. The Ru-based electrocatalysts for HER were reported in previous literatures. In particular, what are the difference and innovation of this paper compared to those (Nat. Nanotech. 2017, 12, 441; Chem. Eur. J. 2019, 25, 8579-8584, Nat. Commun. 2019, 10, 631)?

Response 2.3. In conjunction with the response to the reviewer #1, the reviewer's question is very pertinent. Researches on Ru-based HER catalysts have recently been very active. There are numerous studies as the reviewer indicated (Nat. Nanotechnol. 2017, 12, 441; Chem. Eur. J. 2018, 25, 8579-8584; Nat. Commun. 2019, 10, 631).

Based on these studies, the HER field is now rapidly approaching a commercial stage. There are three important criteria to meet commercial demands, which are cost competitiveness, performance and stability. We focused on developing a catalyst to satisfy these factors as well as suggesting a standard evaluation method for HER performance with a full cell system. We have also enhanced this study by adding EXAFS results (**Supplementary Fig. 1**) and DFT calculations (**Supplementary Fig. 14, 15 and 16**). For more detailed responses to this suggestion, **please see response to the reviewer 1.0**.

Comment 2.4. There are many minor mistakes in the References, I just listed some as below but suggest the authors to check throughout the References and may find more:

1). Page 15, ACS Catalysis 4, 1693-1705 (2014).

Comment: please replace “ACS Catalysis” by “ACS Catal.”

2). Page 15, the Hydrogen Evolution Reaction at All Ph Values. Angew. Chem. Int. Ed. 56, 11559-11564 (2017).

Comment: please replace “All Ph Values” by “All pH Values”

3) Page 15, Mahmood J. et al. An Efficient and Ph-Universal Ruthenium-Based Catalyst for the

Hydrogen Evolution Reaction. Nat. Nanotechnol. 12, 441 (2017).

Comment: please replace “Ph-Universal” by “pH-Universal”.

Response 2.4. Thanks to the reviewer for careful evaluation and pointing out the mistakes. We rechecked the whole references and corrected mistakes.

Reviewer #3 (Remarks to the Author):

Comment 3.1. In this paper, Kweon and his co-workers have reported their work entitled "Ruthenium@carbon nanotube electrocatalyst with superior Faradaic efficiency for hydrogen production". The hybrid sample of ultrathin Ru nanoparticles on MWCNT (Ru@MWCNT) was prepared through a simple reduction method with the aid of sodium borohydride (NaBH₄). Fundamental morphology, structure, catalytic performance and stability of the material in both alkaline and acidic media were studied. Ru@MWCNT displays outstanding HER performance in both acidic and alkaline solutions in almost all aspects such as the overpotential at a current density of 10 mA cm⁻², Tafel slope, the exchange current density, charge transfer resistance, ECSA, and stability, TOF, Faradaic efficiency, as well as various specific activities, etc., obviously superior to benchmark Pt/C. Although the author provides a relatively complete study on the Ru@MWCNT materials for hydrogen evolution in both acidic and alkaline solution, the paper does not provide sufficiently the technical-novelty or scientific achievement in the development of Ru-based HER catalysts. First, the preparation method by the reduction method with the help of NaBH₄ has been widely used in previous works, for instance Adv. Energy Mater. 2018, 1801698, Nature nanotechnology 12.5 (2017): 441.

Response 3.1. First of all, many thanks for taking time to access our manuscript and providing thoughtful feedbacks. The reviewer's comments were helpful to improve the quality of our manuscript.

The uniqueness of our catalyst Ru@MWCNT lies in its simplicity and practicality of synthesis procedure. Ru-based HER catalysts have been actively studied in recent years. Now, the HER field is approaching the commercial stage. There are three important criteria to meet commercial demands, which are cost competitiveness, performance and stability. We focused on developing a catalyst to satisfy these factors. For more detailed responses to this question, please see response to the reviewer 1.0.

We have developed a stable Ru-based catalyst (Ru@MWCNT) using the chemistry between carboxylic acids on MWCNT (MWCNT-COOH) and Ru precursor (Ru³⁺). In this regard, an EXAFS study (**Supplementary Fig. 1**) confirms our assumption on the formation of Ru carboxylate complex. We are not claiming the reduction method using NaBH₄ is new, but it is a general one, which we have also adapted in our previous work (*Nat. Nanotechnol.* 2017, 12, 441-446).

Comment 3.2. Second, only some basic characterization techniques were used which cannot provide insights in the materials. The author employed thiocyanate ions (-SCN) to identify the active sites on the Ru@MWCNT. In order to provide the in-depth understanding about the catalyst for HER, the authors should pay more attention to more advanced characterization techniques such as Operando XPS or X-ray absorption spectroscopy (XAFS) or DFT calculations to reveal the real active sites. Lastly but most importantly, recently, there are many similar literatures about the Ru-C-based materials as HER catalysts in pH-universal electrolytes (*Energy Environ. Sci.*, 2018, 11, 800; *Adv. Energy Mater.* 2018, 1801698; *J. Mater. Chem. A*, 2018, 6, 2311; *Adv. Mater.* 2018, 30, 1803676; *Sustainable Energy Fuels*, 2017, 1, 1028; *J. Mater. Chem. A*, 2017, 5, 25314; *Chem. Commun.*, 2018, 54, 13076; *Catal. Sci. Technol.*, 2017, 7, 4964 etc.). Considering the insufficient novelty of this study, the manuscript was not suitable to be published in Nature Communications.

Response 3.2. Considering the reviewer's opinion, we have conducted DFT calculations and EXAFS analyses to identify the actual active sites, the enhanced catalytic performance and the stability of Ru@MWCNT by calculating the binding energy of Ru-H and the formation energy of Ru@MWCNT (**Supplementary Fig. 14, 15 and 16**). Extended X-ray absorption fine structure (EXAFS) spectroscopy was used to analyze the formation of Ru carboxylate complex and local structural environment of Ru@MWCNT catalyst before and after heat-treatments (**Supplementary Fig. 1**). As a reference, Ru acetylacetonate containing pristine Ru-O bonds was used to confirm Ru-O bonding. The Fourier-transformed (FT) EXAFS spectrum of the reference Ru acetylacetonate shows the major peak at around 1.5 Å, corresponding to Ru-O coordination. Ru@MWCNT before heat-treatment also has Ru-O coordination, which confirms the Ru carboxylate coordination. However, after heat-treatment, Ru@MWCNT shows that the peak at 1.5 Å was slightly shifted to 1.6 Å, indicating the formation of Ru-C coordination. The main peak at 2.4 Å is associated with Ru-Ru coordination in Ru nanoparticles. These results indicate that the formation of Ru carboxylate complexes, which help to form smaller and more uniform Ru nanoparticles during the heat-treatment.

First-principle density functional theory (DFT) calculations were also performed to gain more insight into the enhanced electrocatalytic activity of Ru@MWCNT active sites for HER. It is widely known that the formation energy of metal-hydrogen (M–H) bond plays an important role in hydrogen evolution. Being at the center of volcano plot for electrocatalysts, Pt displays the optimal M-H binding energy, which is neither too weak nor too strong (*Nat. Nanotechnol.* 2017, 12, 441-446). Catalysts having M–H binding energy similar or close to Pt–H (0.53 eV) will efficiently promote hydrogen evolution. The DFT calculations were performed based on our previously reported Ru@C₂N (*Nat. Nanotechnol.* 2017, 12, 441-446) for a clear comparison. To sustain the catalytic activity of Ru nanoparticles, the important point is to protect their aggregation (Ostwald ripening). The calculation showed that Ru nanoparticles on Ru@MWCNT have closer Pt-H binding energy than on Ru@C₂N (**Supplementary Fig. 14**). This result indicates that Ru@MWCNT can have better HER performance than Ru@C₂N. For more details, hydrogen binding energies of possible H binding sites are identified, and the four most stable energies are 0.58, 0.64, 0.64 and 0.62 eV (**Supplementary Fig. 15**). All stable configurations of Ru@MWCNT show lower binding energies than Ru@C₂N (0.68 eV), suggesting that Ru@MWCNT can display enhanced catalytic activity. An important point to be noted that the Ru@MWCNT has an energy of –5.23 eV (10 Ru–C bonds) (**Supplementary Fig. 16**), implying that there are strong Ru-C bonds between Ru nanoparticles and MWCNT. This result reflects the stability of Ru nanoparticles on the surface of MWCNT (Ru@MWCNT) during long cycling test. Furthermore, the formation of Ru–C bonds was confirmed by EXAFS results (**Supplementary Fig. 1**), supporting that the aggregation (Ostwald ripening) of Ru nanoparticles can be hampered by forming strong bonds between Ru and MWCNT.

In addition, we tried to improve the field of HER research from a practical perspective. As the reviewer has pointed out, there are several recent studies on the Ru-C catalysts. Based on these studies, the HER field is getting closer to commercialization.

Many previous reports have amply documented the fundamentals of Ru-based catalysts, including our previous study (*Nat. Nanotechnol.* 2017, 12, 441-446) and others, such as Operando XPS, XAFS or DFT calculations (*J. Am. Chem. Soc.* 2016, 49, 16174-16181; *Energy Environ. Sci.* 2018, 11, 1232-1239; *Nat. Commun.* 2019, 10, 4936). Although the fundamental studies of catalysts are very important in advance, we now believe that practical approach with a full water splitting system is equally important to satisfy commercial standards. This research is focused on developing a practical synthesis and standard evaluation method for commercially viable HER catalyst (hydrogen production per power consumption and Faradaic efficiency) with a full water splitting system (two-electrode system consisting of HER and OER). (**For more details, please see the response to the reviewer 1.0**).

Reviewer #4 (Remarks to the Author):

In the manuscript, the authors claimed a Ruthenium@carbon nanotube electrocatalyst for HER with superior faradaic efficiency. Yet, there are several points that the authors should be noticed clearly.

Comment 4.1. So many Ru based catalysts for HER have been reported. This work doesn't demonstrate their novelty in the material design. As for the synthetic route, this work doesn't show its advantages compared to other noble-metal based catalysts which shown in Supp. Table S2,3.

Response 4.1. We would like to thank the reviewer #4 for the valuable and thoughtful review of our manuscript. As indicated by the reviewer, researches on Ru-based catalysts have recently been very active and many catalysts have been developed via various synthesis routes. Thanks to these active researches, the HER field is now approaching commercial stage. The uniqueness of this work lies in developing a relatively simple and practical catalyst synthesis as well as to provide a standard evaluation method for HER performance (**For more details, please see the response to the reviewer 1.0**).

Comment 4.2. The electrochemical investigation could not satisfy the performance explanation, further theoretical explanation should be involved.

In summary, I think the novelty and quality of this work do not reach the level of Nature Commun.

Response 4.2. Thanks to the reviewer #4 for raising such a pertinent question. As the reviewer pointed out, on one hand, to correlate/explain the performance of the Ru@MWCNT, theoretical study is important. We have added the DFT calculations. The M-H binding energy and formation energy of Ru@MWCNT were calculated to study the activity and stability of Ru@MWCNT. For more detailed discussion, **please also see the response 1.0 (reviewer #1) and 3.2 (reviewer #3)**.

On the other hand, we tried to advance the HER field from a different perspective. With booming researches on Ru based catalysts, plenty of fundamental studies have been made through theoretical calculations such as DFT (*Nat. Nanotechnol.* 2017, 12, 441-446; *Nat Commun.* 2019, 10, 4936). Thanks to these investigations, many excellent Ru-based catalysts have been developed. Theoretical explanation and the role of Ru nanoparticles on carbon support have already been rigorously explored in other literatures (*J. Am. Chem. Soc.* 2016, 49, 16174-16181; *Advanced Materials*, 2018, 30, 1803676; *Energy Environ. Sci.* 2018, 11, 1232-1239; *Nat. Nanotechnol.* 2017, 12, 441-446; *Nat. Commun.* 2019, 10, 4936). The analysis of catalysts based on fundamental chemistry is very important and amply documented, but now practical approach on the actual full water splitting system is equally crucial for commercial stage. The main focus of this manuscript are to present the simplicity and practicality of catalyst synthesis as well as a standard evaluation method for catalyst performance (hydrogen production per power consumption and Faradaic efficiency). For more detailed discussion, **please also see the response to the reviewer 1.0**.

REVIEWERS' COMMENTS:

Reviewer #1 (Remarks to the Author):

The revised version provided by Baek and co-workers clearly improves the original submission and provides pertinent answers to the set of questions raised by the reviewers. Particularly, the authors provide extra characterization data (EXAFS) of the formed Ru(III) complexes at the surface of MWCNTs prior to NP formation and include new Ru-H adsorption energies from DFT calculations that support the excellent HER activity of the supported system when compared with their previous work in Nat. Nanotechnol. (2017). Also, a fair comparison of the most relevant figures of merit (TOF, overpotential) is now included in Figure 3d. In this regard, I would suggest including appropriate references from where these numerous data have been collected (recent reviews). All in all, even if the synthetic methodology does not provide novel concepts to the field, the top performance of the catalyst in HER and its performance in the reported overall WS device are sound enough to merit publication in Nature Communications.

Reviewer #2 (Remarks to the Author):

The authors have addressed all my concerns and thus I would suggest the acceptance of this manuscript as is.

Reviewer #3 (Remarks to the Author):

The authors addressed the comments to some extent and revised the paper accordingly. The revised paper could be considered for publication.